# Leveraging deep learning for comprehensive classification of renal diseases: a transfer learning approach

## Abstract

The nightmare of cancer as a leading cause of premature deaths worldwide is becoming real and turns out to be one of the major problems of humanity nowadays. Cancer diagnostics at the early stage is Critical to cancer recovery and survival. In this context, renal diseases, including kidney cysts, stones, and tumors, pose significant global health challenges, affecting approximately 12% of the population and contributing to chronic kidney disease (CKD). Notably, renal cancer ranks as the tenth most prevalent cancer type, accounting for 2.7% of all cancer cases. This work presents a deep learning (DL) framework utilizing transfer learning (TL) for the early detection of renal diseases and categorizing the conditions into four binary classifications: Cyst_vs_Normal, Cyst_vs_Stone, Cyst_vs_Tumor, and Stone_vs_Tumor, allowing for a more specific understanding of each stage. By analyzing CT scans and microscopic histopathology images, the framework employs convolutional neural networks (CNNs) with pre-trained models to facilitate automatic and precise classification of renal conditions. Specifically, two CNN models ResNet-50 and EfficientNetV2 are implemented, providing a comprehensive analysis of each stage of the DL architecture. Comparative evaluations of training outcomes across various datasets revealed that EfficientNetV2 performed marginally better than ResNet-50, achieving an impressive testing accuracy of up to 100% for all cases. These results underscore the effectiveness of the DL-based system and highlight its potential for widespread clinical application in renal disease diagnosis.

Keywords- CNN, kidney, image classification, deep learning, transfer learning

## 1 Introduction

Renal organs are vital, bean-shaped located below the rib cage that filters blood, removes waste, and balances fluids and electrolytes, with each kidney containing about a million nephrons Raghavendra & Vidya (2013). Additionally, they control blood pressure, stimulate red blood cell production through hormone secretion, and maintain overall homeostasis, making their proper function essential for health. Renal cancer originates in the kidneys when malignant cells form in the tubules, often requiring treatments like immunotherapy Navani & Heng (2023). Early detection and advanced medical interventions can significantly improve outcomes for affected individuals Jacobson (2013). Kidney cancer is a growing public health concern, in the year 2022, it ranked as the 14th most common cancer and the 16th leading cause of cancer-related death globally Can (2022) to the 5th by 2040 Foreman et al. (2018). Kidney diseases commonly consist of cysts, stones, and renal cell carcinoma (RCC) Hsieh et al. (2017), while nephrolithiasis affects approximately 12% of the global population Alelign & Petros (2018). Despite control efforts, the prevalence of these conditions continues to increase, highlighting the need for enhanced medical interventions and public health strategies Hsieh et al. (2017). Computed tomography (CT) scans are particularly effective for kidney examinations, offering three-dimensional, cross-sectional images ideal for identifying abnormalities like cysts, stones, and tumors Sagel et al. (1977).

It is challenging for doctors to clinically determine invasive cancer by identifying the captured images, and it needs to administer safe and expensive treatments for the patients Wang et al. (2020). Therefore, tracking the growth of every stage is essential to develop customized medications based

on a patient's disease profile Hsieh et al. (2017). There is a significant global shortage of nephrologists and radiologists, particularly in Asia, where there is only about one nephrologist for every million people, in contrast to Europe, which has approximately 25.3 nephrologists per million Sozio et al. (2021). Given the widespread impact of kidney diseases and the scarcity of specialists, developing DL-based models to assist in detecting kidney abnormalities has become essential Bi et al. (2022). Recent advancements in DL-based models for vision tasks offer promising solutions to support doctors and alleviate patient suffering.

This work is organized as follows: Section 2 discusses recent work that has been done previously. Section 3 provides an overview of the dataset and details of the model training process, including the initialization of weights. Additionally, this section offers a comprehensive overview of the DL architectures employed. Two different CNN models are introduced, evaluating their performance for detection across the four stages of renal disease. Finally, Section 4 presents concluding remarks and outlines potential pathways for future works.

## 2 RELATED WORKS

CNNs are a powerful DL algorithm commonly used for classifying grid-like data, such as images Simonyan & Zisserman (2015); Szegedy et al. (2014); He et al. (2015a); Tan & Le (2020). Specifically, renal US images are enhanced using median and Gaussian filtering techniques and morphological operations Verma et al. (2017). Relevant features from images are extracted using various unsupervised techniques and classified using supervised algorithms. In Aksakalli et al. (2021), authors demonstrated a range of traditional supervised machine learning algorithms, including decision tree (DT), random forests (RF), K-nearest neighbors (KNN), and multilayer perceptron (MLP), as well as CNN. They achieved the best F1 score of 0.853 with those methods. In Sudharson & Kokil (2020), they employed pre-trained CNN models like ResNet-101, MobileNet-v2, and ShuffleNet to extract features from kidney US images, got an accuracy of 95.58% using support vector machine (SVM). In Fu et al. (2021) residual dual-attention (RDA) module utilized for segmenting kidney cysts from the CT images. In Zheng et al. (2019), combined features extracted through TL approaches, which were subsequently utilized to distinguish between affected and non-affected ultrasound images by SVM classifier. In Parakh et al. (2019), two consecutive CNN models were employed: first CNN identified the urinary region, and second CNN detected the existence of stones, both achieved an accuracy of 95%. In Yildirim et al. (2021) introduced an automated method for the detection of kidney stones using coronal CT images and DL techniques, achieving an accuracy of 96.82%. In Blau et al. (2018), researchers developed a system to detect kidney cysts in the images of abdominal CT scans by utilizing a fully connected CNN. They reported an 84.3% true-positive rate (TPR) for their approach. In recent studies on kidney disease detection from CT images, the EANet, ResNet50, and a customized CNN model achieved accuracies of 83.65%, 87.92%, and 98.66%, respectively Hossain et al. (2023). In summary, initiatives employing ML and DL approaches to classify kidney-related radiological findings have demonstrated encouraging outcomes, primarily concentrating on CT and US images.

The rapid advancement of CNNs has led to the development and utilization of various DL architectures Khan et al. (2020), including EfficientNet Tan & Le (2020) a highly efficient CNN by Google AI that balances depth, width, and resolution for superior performance and reduced computational cost in image classification tasks. After that, it was updated utilizing sophisticated DL methods like the fused mobile inverted bottleneck (Fused-MBConv) operation to produce even better performance and was named EfficientNetV2 Tan & Le (2021). While ResNet-50 CNN model has been granted as a promising CNN model for image classification task He et al. (2015b). This study utilized the EfficientNetV2 architecture and contrasted its performance with ResNet-50 for the classification of four different stages of kidney cancer image dataset "CT KIDNEY DATASET" Kid (2022). This study subdivided into four binary classifications: Cyst_vs_Normal, Cyst_vs_Stone, Cyst_vs_Tumor, and Stone_vs_Tumor, aiming to distinguish between specific kidney conditions and streamline the diagnostic process. Each of them trained to differentiate between them, allowing for more focused and precise detection of kidney abnormalities. Those CNN models demonstrate high accuracy across all conditions, highlighting their potential for kidney cancer diagnosis and prediction.

Table 1: Kidney images dataset summary at four different stages

| Label/Set | Total Count | Training | Validation | Test |
|---|---|---|---|---|
| **Normal vs Cyst** | | | | |
| Normal | 5077 | 4077 | 479 | 521 |
| Cyst | 3709 | 2951 | 399 | 359 |
| *Combined* | 8786 | 7028 | 878 | 880 |
| **Cyst vs Stone** | | | | |
| Cyst | 3709 | 2992 | 349 | 368 |
| Stone | 1377 | 1077 | 158 | 142 |
| *Combined* | 5086 | 4069 | 507 | 510 |
| **Cyst vs Tumor** | | | | |
| Cyst | 3709 | 2993 | 359 | 357 |
| Tumor | 2283 | 1800 | 240 | 243 |
| *Combined* | 5992 | 4793 | 599 | 600 |
| **Stone vs Tumor** | | | | |
| Stone | 1377 | 1084 | 152 | 141 |
| Tumor | 2283 | 1844 | 214 | 225 |
| *Combined* | 3660 | 2928 | 366 | 366 |

## 3 RESULT AND DISCUSSION

### 3.1 DATSET OVERVIEW

The uploaded dataset in Kaggle Kid (2022), primarily sourced from picture archiving and communication system (PACS) at Dhaka, Bangladesh hospital Sharma & Lalwani (2024). Contains both axial cuts and coronal from disparity and non-disparity studies, following protocols for the entire abdomen and urogram, which were selected for collecting the images. Figure 1 displays sample images of kidneys, with red marks highlighting the regions of interest that radiologists use to make specific diagnoses.

| Normal | Cyst | Stone | Tumor |
|---|---|---|---|

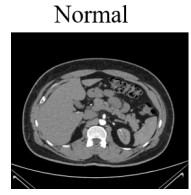 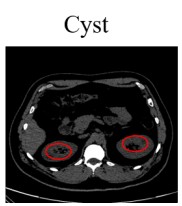 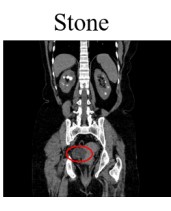 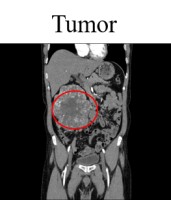

Figure 1: Sample CT scans kidney cancer image data Kid (2022). (a) A normal kidney with, a consistent and uniform structure without any abnormalities, serves as a baseline for comparison with the other conditions. (b) Cysts, types of sacs filled with fluid, may vary in size and can sometimes cause pain or other complications. (c) Stone, a hard deposit made of minerals and salts, can cause severe pain and may require medical intervention. (d) A tumor, an abnormal growth of tissue, can be benign or malignant (cancerous), such as RCC. The highlighted areas in red indicate the presence of abnormal growth of tissue

Table 1 presents a dataset summary of the four conditions, detailing the total number of images, in addition to the number of images for training, testing, and validation respectively. The "Normal_vs_Cyst" dataset at the top contains the largest number of images, totaling 8786 for training, validation, and testing purposes. The "Cyst_vs_Stone" dataset in the middle comprises 5086 images, "Cyst_vs_Tumor" dataset below it includes 5992 images. Lastly, the "Stone_vs_Tumor" dataset at the bottom consists of 3360 images.

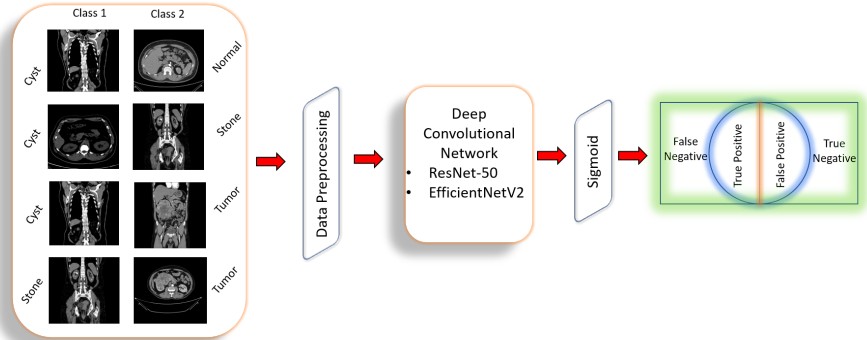

Figure 2: Complete Workflow of an experiment for diagnosing kidney cancer at different conditions. Images are resized to 224×224 as a height and width. After preprocessing the data, deep CNNs He et al. (2015b), Tan & Le (2021) are then developed and trained, with weights optimized through gradient descent. Sigmoid layer outputs provide the score for classifying each image. Finally, the accuracy metric is used to evaluate the model's performance.

## 3.2 INITIALIZATION OF WEIGHTS

Weight initialization is a critical factor influencing the efficiency and effectiveness of NN training Vedanshu & Tripathi (2018). When weights are initialized to zero, a symmetry problem arises during gradient descent, causing neurons within a layer to learn identical features and hindering effective learning. To address this, specialized initialization methods like Xavier Glorot & Bengio (2010) and He He et al. (2015c) initialization are utilized, each tailored to different activation functions. Xavier initialization, also known as Glorot initialization Glorot & Bengio (2010), is designed for layers using activation functions such as tanh and sigmoid. This method initializes weights using a uniform distribution:

$$W_{\text{Xav}} = U\left(-\frac{1}{\sqrt{m}}, \frac{1}{\sqrt{m}}\right) \tag{1}$$

Where the number of input neurons is represented by *m*. Xavier initialization helps maintain the variance of activations and gradients throughout the network layers Ramachandran et al. (2017). This promotes smoother convergence during gradient descent, preventing the gradients from vanishing or exploding, which can be crucial for deep networks. For activation functions like ReLU or SiLU Agarap (2019), Xavier initialization can lead to vanishing gradients, particularly for deep networks. He initialization, also known as Kaiming initialization, addresses this by using a Gaussian distribution for weight initialization:

$$W_{\text{He}} = \mathcal{N}\left(0, \sqrt{\frac{2}{m}}\right) \tag{2}$$

where $\mathcal{N}(0, \sigma^2)$ denotes a normal distribution with mean 0 and $\sigma^2$ He et al. (2015c). This approach scales the weights to ensure that the variance of the activations remains consistent, which is particularly important for ReLU and similar activation functions that can otherwise suffer from sparse gradients. Modern NN architectures such as ResNet-50 and EfficientNetV2 predominantly use ReLU or SiLU activation functions, as a result, creators of those models used He initialization in these layers He et al. (2015b). Consequently, these models typically employ He initialization to ensure effective training. ResNet-50, leverages residual connections that help mitigate the vanishing gradient problem, and He initialization further complements this by maintaining appropriate gradient scales. EfficientNetV2, emphasizing optimization accuracy and computational efficiency, similarly benefits from He initialization to achieve robust performance.

TL Zhuang et al. (2020) is a powerful technique that significantly enhances training efficiency, especially when data is limited. It involves taking a pre-trained model, often trained on a large dataset like ImageNet, and fine-tuning it for a different but related task. By initializing the model with pre-trained weights, such as those from ImageNet Abadi et al. (2016b), TL allows the model to

leverage previously learned features and patterns. This provides a strong starting point and leads to more effective training with less data Abadi et al. (2016b). For example, when fine-tuning ResNet-50 or EfficientNetV2 for a new task, initializing with pre-trained ImageNet weights enables faster convergence and often results in better performance than training from scratch.

## 3.3 TRAINING RESULT AND DL MODELS

Two modern CNN architectures ResNet-50 and EfficientNetV2 are used, and both models were trained and evaluated using TensorFlow Abadi et al. (2016b); Tan & Le (2021); He et al. (2016). To ensure optimal performance, a TensorFlow checkpoint mechanism was implemented for tracking the accuracy of each epoch during the validation process, which allowed us to keep into record of the topmost model weights after the training process Abadi et al. (2016a). The training for both ResNet-50 and EfficientNetV2 was conducted over 10 epochs with 256 iterations per epoch. The best weights for the ResNet-50 model were saved at epochs 06, 10, 05, and 08 for the Cyst_vs_Normal, Cyst_vs_Stone, Cyst_vs_Tumor, and Stone_vs_Tumor datasets, respectively. In contrast, the EfficientNetV2 model achieved its best weights at epochs 09, 10, 07, and 10 for the same datasets.

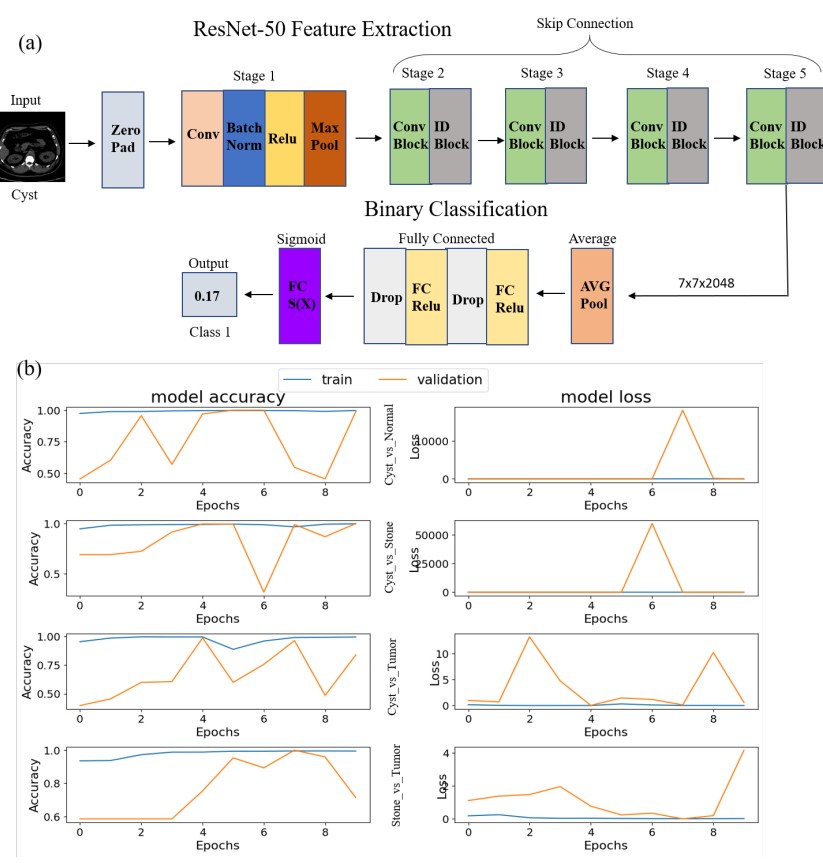

Figure 3: (a) A brief architecture of the Resnet-50 model representing through step-by-step process. The framework demonstrates a forward-passing process through a feature extraction stage (at the top) and a binary classification stage (at the bottom). (b) Graphical representation of training for four conditions. Left and right column represents the model accuracy and loss after the training (blue line) and validation (orange line) process.

The ResNet-50 model, developed by Microsoft in 2015, is a well-known CNN architecture demonstrating outstanding accomplishment on the ImageNet dataset He et al. (2015b). As shown in Figure 3 (a), the architecture comprises a feature extraction and a binary classification. At the top five distinct stages are included for feature extraction, which recognize the most valuable features from the given input image tensor, Apart from that binary classification layers are employed at the bottom

using the sigmoid activation function S(x). During feature extraction, the last four layers (2–5) contain a combination of convolutional and identity blocks He et al. (2015b), Simonyan & Zisserman (2015), as depicted in Figure 3 (a). Each of these blocks incorporates a skip connection, in addition to convolutional layers and batch normalization Ioffe & Szegedy (2015). These skip connections help mitigate issues related to vanishing and exploding gradients, allowing the model to effectively utilize a greater number of layers and learn more complex features, ultimately leading to improved accuracy. After extracting high-level features, the intermediate output image tensor got flattened before passing through the final fully connected layer as illustrated in Figure 3 (a). The feature extractor's output (originally 7 × 7 × 2048) is flattened out into a one-dimensional array of size 2048, by averaging the first two dimensions using global average pooling Lin et al. (2014). FC and dropout layers then reduce the 2048-node array to a one-dimensional final output of 512 nodes. The final prediction is obtained by the FC layer at the end, which includes a sigmoid activation function.

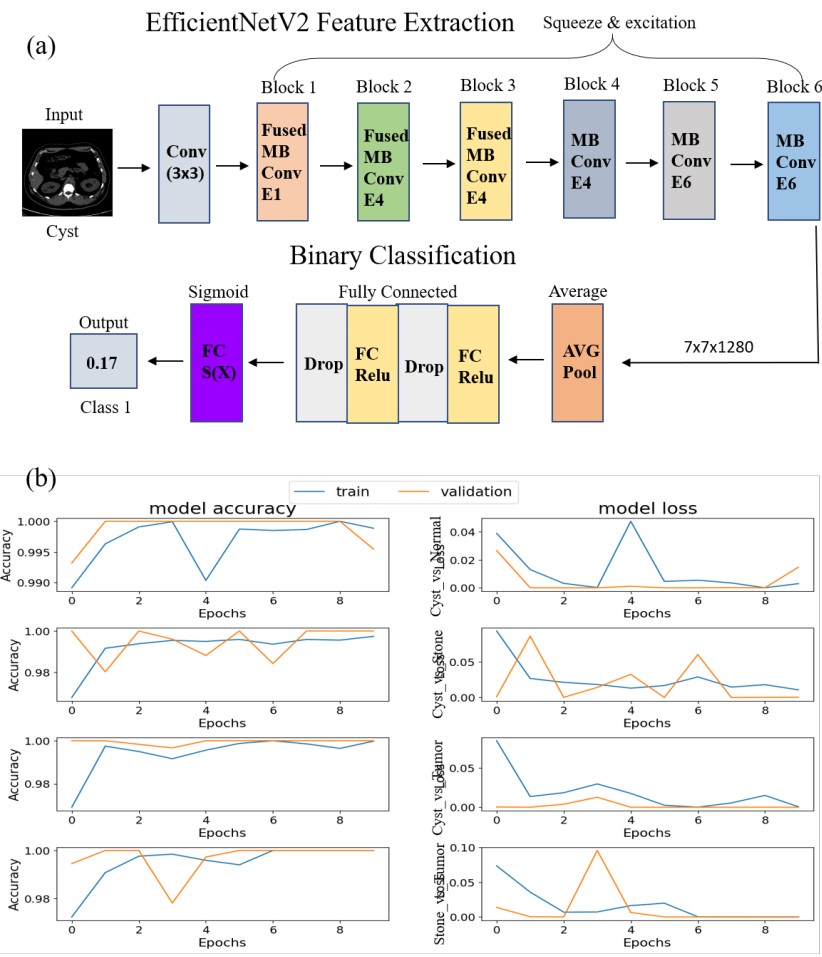

Figure 4: (a) A brief architecture of the EfficientNetV2 model representing through step-by-step process. The framework demonstrates a forward-passing process through a feature extraction stage (at the top) and a binary classification stage (at the bottom). The number of channels at each MB-Conv block is expanded, represented by the expansion component (E). (b) Graphical representation of training for four categories. Left and right column represents the model accuracy and loss after the training (blue line) and validation (orange line) process.

The training and validation process results, shown in Figure 3 (b), demonstrate the ResNet-50 model accuracy and loss for each category over the epochs. The training accuracy and loss consistently trend upward and downward from the start, while validation shows some fluctuations. Implementing a TensorFlow checkpoint Abadi et al. (2016b), the most effective epoch is identified throughout the training process. As a result, the best weights were retained corresponding to the best validation

accuracy, ensuring that the validation images were not included in the training phase. This approach led to improved accuracy in testing.

Now for the second model, EfficientNetV2 represents cutting-edge supervised tasks, developed by Google Brain in 2021 Tan & Le (2021). This model leverages the method, known as neural architecture search (NAS) Tan et al. (2019), which systematically explores various ML architectures to identify the most effective design by sampling different configurations and evaluating their performance. EfficientNetV2 builds upon its predecessor, EfficientNetV1, by optimizing speed during training and incorporating a valuable operation, Fused-MBConv at the earlier layers. Unlike the original EfficientNetV1, which employs depthwise convolutions, the Fused-MBConv layer utilizes standard $3 \times 3$ convolutions, as illustrated in Figure 4 (a). Additionally, EfficientNetV2 incorporated strategies from several prior studies to enhance training efficiency keeping the number of parameters at manageable levels. One such technique is progressively adaptive regularization by learning, providing a regularized framework that aligns with the image resolution during training Sandler et al. (2019). This approach gradually adjusts the regularization and image size, starting with lower values in the early epochs and increasing them in later stages. Although progressive learning was not implemented here, the TL methods reduced the number of parameters and allowed for a significant reduction in training time approximately less than the ResNet-50 model. The principal components of the EfficientNetV2 architecture include MBConv and Fused-MBConv layers, which are reused multiple times throughout the model. As depicted in Figure 4 (a), the architecture is divided into segments for feature extraction and binary classification similar to ResNet-50. The feature extractor begins with a stem that includes a standard convolution layer, followed by six distinct blocks that consist of several repetitions of Fused-MBConv and MBConv layers. After collecting feature tensors of size $7 \times 7 \times 1280$, global average pooling is applied Lin et al. (2014), resulting in a single value per channel. Then passing through three layers (two ReLU activation and one sigmoid activation) output is obtained.

The training and validation process results, shown in Figure 4 (b) demonstrate the model accuracy and loss over each category across the epochs. The Cyst_vs_Normal dataset is shown at the top, followed by Cyst_vs_Stone and Cyst_vs_Tumor in the middle, and Stone_vs_Tumor at the bottom. Similar to ResNet-50 the most effective epoch is identified throughout the training process, by implementing a TensorFlow checkpoint Abadi et al. (2016b). As a result, the best weights are retained corresponding to the best validation accuracy. While the training time of EfficientNetV2 is much faster than the ResNet-50. Overall, the EfficientNetV2 model demonstrates comparable stability in accuracy and loss to the ResNet-50 model. Training with EfficientNetV2 was approximately faster than ResNet-50, thanks to optimizations introduced through neural architecture search (NAS).

### 3.4 MODEL PERFORMANCE

The performance of the ResNet-50 model across four binary classifications demonstrates its high reliability in medical image classification [see Figure 5]. The model achieves perfect or near-perfect accuracy in all four conditions, with minimal misclassifications, highlighting its robust capability in distinguishing between the conditions. The Cyst_vs_Tumor task shows a slight decrease in performance compared to the others, reflecting the inherent challenge in differentiating these two classes. Precision remains consistently at 100% for all tasks, indicating the absence of false positives, while recall is also perfect except for a minor reduction in the Cyst_vs_Tumor task. These results establish ResNet-50 as a highly effective tool for binary classification in medical imaging, capable of delivering reliable predictions with minimal errors across diverse categories.

In contrast, the EfficientNetV2 model demonstrates strong performance across the four binary classification tasks [see Figure 6]. It achieves perfect accuracy for all tasks and near-perfect accuracy in the Cyst_vs_Stone. Precision remains consistently high across all of the classifications, with only a slight drop in the Cyst_vs_Stone. Similarly, recall is perfect across all classifications. These results underscore the model's reliability and effectiveness in medical image classification, with minimal errors, making it well-suited for clinical applications.

Both models exhibit similarly high performance across all four binary classification tasks, but EfficientNetV2 shows a slight advantage in the Cyst_vs_Tumor, where it achieves flawless classification, unlike ResNet-50, which had minor misclassifications. However, EfficientNetV2 experienced a small dip in precision during the Cyst_vs_Stone task, where ResNet-50 maintained perfect precision.

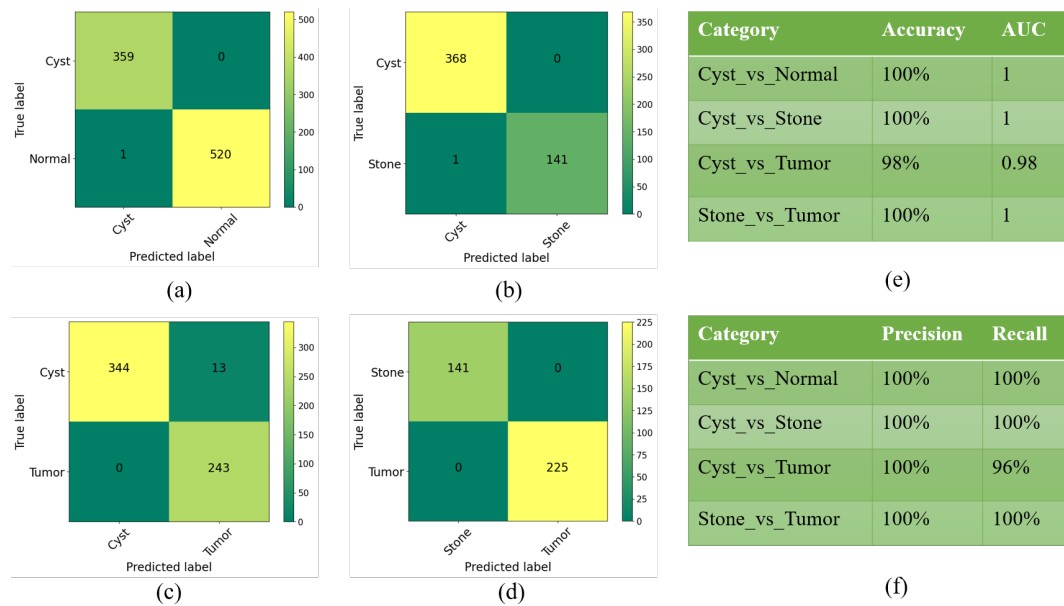

Figure 5: Performance of ResNet-50 model over the test sets of four different conditions. (a) Cyst_vs_Normal (b) Cyst_vs_Stone (c) Cyst_vs_Tumor (d) Stone_vs_Tumor represents binary confusion matrix. (e) shows the accuracy and AUC score for each binary classification, while (f) displays the corresponding precision and recall metrics.

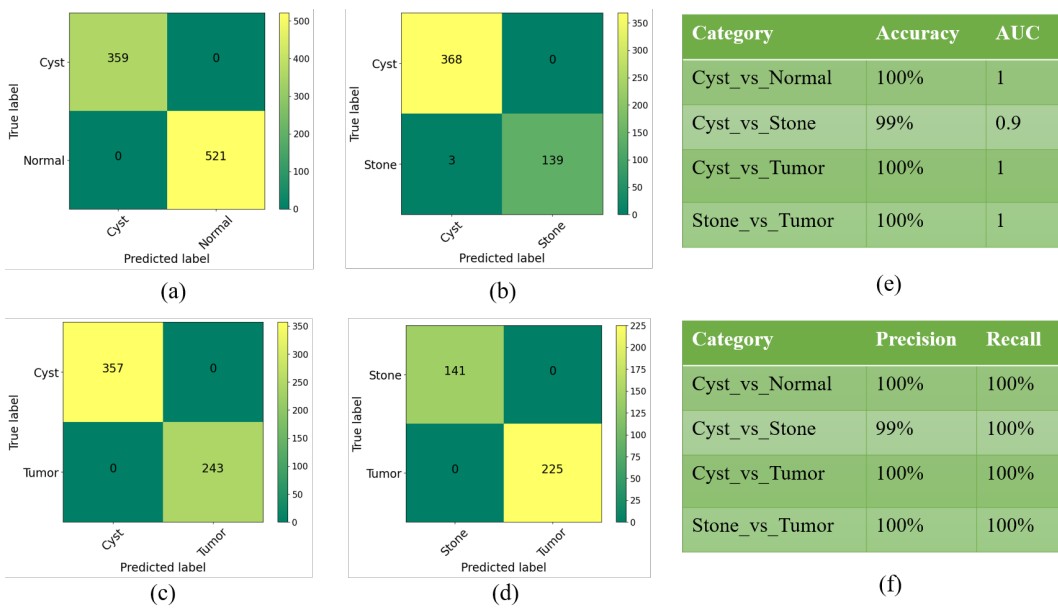

Figure 6: Performance of EfficientNetV2 model over the test set. (a) Cyst_vs_Normal (b) Cyst_vs_Stone (c) Cyst_vs_Tumor (d) Stone_vs_Tumor represents binary confusion matrix. (e) shows the accuracy AUC score for each binary classification, while panel (f) displays the corresponding precision and recall metrics.

Overall, both models deliver remarkable accuracy, EfficientNetV2 shows a slightly more consistent recall performance across all tasks, making it a marginally more robust model for medical image classification, especially when subtle class distinctions are involved. Even small gains in accuracy,

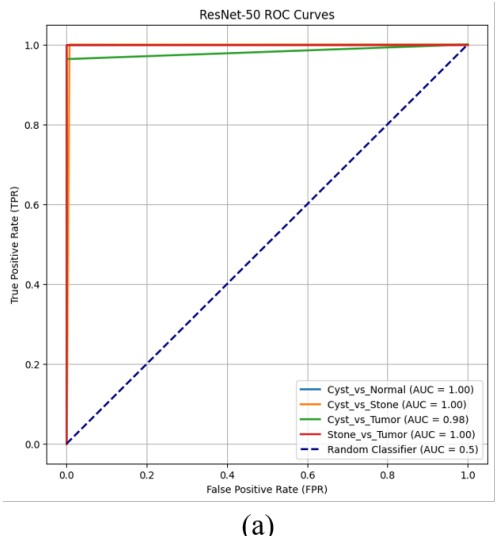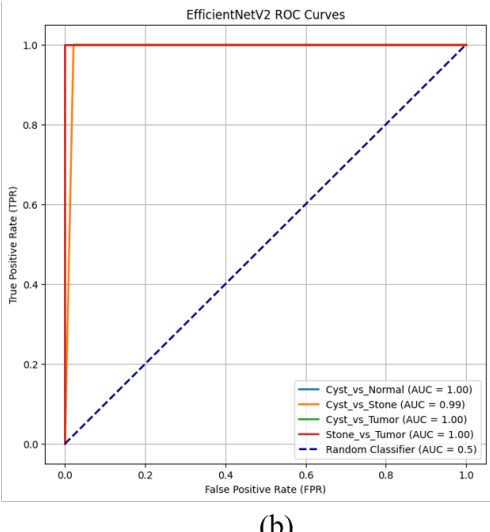

(a)                  (b)

Figure 7: Receiver Operating Characteristic (ROC) curves for two deep CNN models: (a) ResNet-50 and (b) EfficientNetV2, across multiple binary classification tasks. The plots represent the True Positive Rate (TPR) against the False Positive Rate (FPR) for the four different conditions: Cyst_vs_Normal, Cyst_vs_Stone, Cyst_vs_Tumor, and Stone_vs_Tumor, along with a reference random classifier (AUC = 0.5). The AUC values indicate excellent classification performance, with most tasks achieving AUC values close to 1.

as seen with models like EfficientNetV2 compared to ResNet-50, could have significant implications for future medical applications. The high levels of accuracy achieved with these datasets highlight the potential of deep CNNs in medical image analysis. These findings point toward a promising research direction in the precise identification of renal disease subtypes, where the ability of CNNs to detect subtle variations in images could lead to more effective and reliable diagnostic tools in healthcare.

## 4 CONCLUSION

The DL frameworks presented in this study effectively address the critical need for early detection of renal diseases. By leveraging TL and advanced CNNs system achieved remarkable testing accuracy of up to 100% across multiple classifications. These findings not only demonstrate the potential for precise and automatic classification of renal conditions but also highlight the framework's applicability in clinical settings. By improving early diagnosis, this approach could lead to better patient management, reduced rates of chronic kidney disease, and ultimately improved survival rates in renal cancer. However, future work is needed to inflate the robustness and generalizability of the model. This includes expanding the dataset to encompass a wider variety of demographic and clinical variables, integrating multimodal imaging techniques, and conducting prospective clinical trials to validate performance in real-world scenarios. Additionally, further research could focus on the interpretability of the model's predictions to support clinicians in decision-making processes.

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
