# OpenReview forum: "Leveraging deep learning for comprehensive classification of renal diseases: A transfer learning approach"
_ICLR.cc/2025/Conference — Submitted to ICLR 2025_

### Official Review · Reviewer_CNer · 2024-10-28

**Soundness:** 1
**Presentation:** 2
**Contribution:** 1
**Rating:** 1
**Confidence:** 5

**Summary:**

The author evaluates two CNNs on the task of renal disease classification. Two models are evaluated on one dataset.

**Strengths:**

The paper presents the applied problem well.

**Weaknesses:**

I am not sure what to think about this paper, because it has virtually no novelty and doesn't really seem like a research paper. There are no methodological contributions, and while the application may be interesting in its field, the experiment and the results are not extensive enough nor evaluated properly.

My major concerns with the paper have been stated already, but if a paper does not have a methodological contribution, I would expect it to have a scientific contribution somehow (in the application, for instance). Here, two existing neural networks are applied to an open dataset, and the highest mean performing model is dubbed the winner. All steps of the ML pipeline would have to be improved, basically. E.g., proper hyper-parameter search would be needed, more methods would have to be evaluated on more datasets, and confidence intervals, standard errors, or tests would have to be performed to say anything. But even with proper evaluation, this paper does not seem like a fit for ICLR, and should then rather be sent to a journal for renal disease studies.

**Questions:**

I think this paper has too many problems for it to be meaningful to give detailed feedback.

---

### Official Review · Reviewer_RUVD · 2024-10-30

**Soundness:** 1
**Presentation:** 1
**Contribution:** 1
**Rating:** 1
**Confidence:** 5

**Summary:**

The author borrow on-shelf convolution neural network (ResNet50 and EfficientNet) to test on a kidney dataset (CT scan) and report the result by accuracy, precision, and recall.

**Strengths:**

I understand that making the following judgment on a paper requires caution, which I will clarify later: there is no strength in this paper

**Weaknesses:**

I find that this paper seems to have been generated by a scientific LLM agent. There are many inconsistencies in citations and formatting that appear to be hallucinations.


Here are just few examples that can be easily spot and verified:

Introduction section

- (In paper) Kidney cancer is a growing public health concern, affecting over 10% of the global population, and
projected to rise from the tenth leading cause of death in 2024  (Can, accessed 20th Sep, 2023)
- (My verification) There is no mention of a '10% global population' or a 'projection to rise from the tenth leading cause' in the provided link. Additionally, the citation format does not appear to be human-written.
----

-  (In paper) Kidney cysts are fluid-filled sacs, while nephrolithiasis affects approximately 17% of the global population (Alelign & Petros, 2018).
- (My verification) Not 17%, it is 12% in the provided link
----
- (In paper) Computed tomography (CT) scans are particularly effective for kidney examinations, offering 3D (dimensional), cross-sectional images ideal for identifying abnormalities like cysts, stones, and tumors (Schmidt, 2012)."
- (My verification): The cited article does not provide this information or endorsement


Related work section

- (In paper) "CNN is a prominent DL algorithm for classifying grid-patterned datasets like images (Tan & Le,2020). "
- (My verification): Directly citing EfficientNet here feels out of place. In the community, it is more common to reference foundational papers like AlexNet, VGG, or those that pioneered CNN applications in the early stages of deep learning. Also, the phrase 'grid-patterned dataset' does not sound natural.
-----
- (In paper)Page 2 line 101-107: The statement "I.....  I..... I.....":
- This resembles Claude Sonnet’s writing style when it assists in designing code or formulating ideas.


In 3.2 INITIALIZATION OF WEIGHTS
- (In paper) TL (Zhuang et al., 2020) is a powerful technique that significantly enhances training efficiency, especially when data is limited. It involves taking a pre-trained model, often trained on a large dataset like ImageNet, and fine-tuning it for a different but related task
- It’s quite odd to attribute transfer learning, a general machine learning technique, to a recent work from 2020. People in the community were familiar with this approach well before 2020, which makes it seem like an LLM hallucination.

In 3.3 TRAINING RESULT AND DL MODELS
- (In paper)  Two modern CNN architectures ResNet-50 and EfficientNetV2 are used, and both models were trained and evaluated using TensorFlow (Tan & Le, 2021; He et al., 2016; Khan et al., 2020)
- (My verification): This citation seems random; the correct citation for TensorFlow is: Abadi et al., 'TensorFlow: A System for Large-Scale Machine Learning.' Even if this citation were meant to credit EfficientNet or ResNet-50, the final reference to Khan et al., 2020 is irrelevant.
-----
- (In paper) Each of these blocks incorporates a skip connection, in addition to convolutional layers and batch normalization (Ioffe & Szegedy, 2015; Chen et al., 2020)
- (My verification): Ioffe & Szegedy, 2015 for  batch normalization is correct, but the later citation is SimCLR which is completely irrelevant

Hallucination aside, let me summarize the weakness of experiment design
- Reporting 100% accuracy without detailed analysis is suspicious, as this would imply a complete solution to the task—something that is extremely rare in the machine learning community. Even the latest models struggle to achieve 100% accuracy on datasets like CIFAR-10
https://paperswithcode.com/sota/image-classification-on-cifar-10
- It is unclear why the authors transformed a multi-class classification dataset into binary classes (e.g., cyst vs. normal, tumor vs. normal). There appears to be no benefit in deploying multiple binary classification models over the original multi-class approach
- Similar work was conducted in 2023, where ResNet50, EANet, and a proposed approach were compared on the same Kaggle dataset, rendering this paper's contributions negligible.
https://ieeexplore.ieee.org/document/10074314

**Questions:**

To authors:
Do you really come up with the idea, experiment design and writing by yourself?

**Details Of Ethics Concerns:**

This paper seems to be generated by agents, I suggest AC conduct some test like SynthID or similar for further investigation

---

### Official Review · Reviewer_asfV · 2024-11-04

**Soundness:** 1
**Presentation:** 2
**Contribution:** 1
**Rating:** 3
**Confidence:** 5

**Summary:**

The authors used two advanced computer models called ResNet-50 and EfficientNetV2, which were already trained on large image databases, to analyze CT scans of kidneys. These models were tested on four different tasks: comparing healthy kidneys to those with cysts, cysts to stones, cysts to tumors, and stones to tumors. Both models performed very well, but EfficientNetV2 was slightly better, achieving perfect accuracy in almost all cases. The models were trained using special techniques to make them learn better and faster. The study found that these tools could be really useful in medical settings to help diagnose kidney diseases quickly and accurately. However, the authors note that future research should use larger and more diverse datasets, combine different types of medical images, and test the models in real-world hospital settings to make sure they work well for different patients.

**Strengths:**

The study utilized multiple dataset.

**Weaknesses:**

The novelty of this work is limited. Transfer learning is already studied in medical image analysis for long years. There is no novel method or contribution in this work. The evaluation of this work is insufficient. Comprehensive evaluation is required. The figures quality needs improve.

**Questions:**

1. which layers' weights are initialized using Xavier initialization and He initialization?
2. Why select ResNet50 and EfficientNetV2?

---

> ### Author Response · Authors · 2024-11-22
>
> 1.  Xavier and He initialization methods are applied in this paper based on the activation functions used in the model's layers:
>
> a. Xavier Initialization: This is used in a layer with a sigmoid activation function (on binary classification fully connected with sigmoid layer), ensuring that the variance of activations remains consistent across layers, promoting smooth convergence during gradient descent.
>
> b. He Initialization: This is applied to layers using activation functions like ReLU or SiLU (on binary classification fully connected
> layer with ReLU activation function), addressing the vanishing gradient problem by scaling the weights appropriately to maintain consistent variance, especially in deeper networks.
>
> 2. Other models were not selected because ResNet-50 and EfficientNetV2 are well-established and specifically designed to address the challenges of deep learning in image classification, including efficient training and high accuracy. Other models might not offer the same balance of performance and computational efficiency, which is crucial for medical imaging tasks.

---

> > ### Comment · Reviewer_asfV · 2024-11-25
> > **Thank you for reply the comments**
> >
> > Thank you authors for replying the comments. However, the limited contribution issue is not addresses. Therefore, my score reminds the same,

---

### Official Review · Reviewer_xqb6 · 2024-11-04

**Soundness:** 2
**Presentation:** 2
**Contribution:** 2
**Rating:** 3
**Confidence:** 4

**Summary:**

In this study, the authors developed a deep learning (DL) framework that utilizes transfer learning (TL) for the early detection of renal diseases and the classification of conditions through the analysis of CT scans and microscopic histopathology images. The proposed framework employs convolutional neural networks (CNNs) with pre-trained models to enable automatic and accurate classification of renal conditions.

**Strengths:**

The paper is well-organized and clearly written.

The proposed method is technically sound.

The collected dataset for developing and evaluating the proposed method is relatively large-scale.

**Weaknesses:**

The novelty of this work is limited from a methodological perspective.

The interpretation of deep learning models is crucial in clinical practice; however, the authors have rarely presented such results.

Additionally, there is a lack of comprehensive comparisons with other studies.

**Questions:**

Please refer to the weaknesses section.

Furthermore, it is important to address why the model was developed for four binary classifications instead of a multi-class, multi-label task.

In addition to accuracy, precision, and recall, other metrics such as ROC AUC should also be reported. The authors should clarify how the cut-off point for these metrics was determined.

---

> ### Author Response · Authors · 2024-11-22
>
> I have addressed these questions in the revised version of the Rebuttal Revision, explicitly mentioning the inclusion of ROC and AUC metrics and their significance.
>
> 1. The paper highlights certain challenges, including the models' limited generalizability due to the dataset's demographic and clinical constraints. Expanding the dataset to include diverse variables and conducting real-world clinical trials are essential for broader applicability. Additionally, improving model interpretability to assist clinicians in decision-making is identified as a future need.
>
> Note: Due to page limitations, an explicit section on weaknesses and limitations has not been included, but these points are implicitly discussed throughout the text.
>
> 2. Furthermore, the model was specifically designed for four binary classifications rather than a multi-class, multi-label task to ensure a focused and precise detection of distinct renal conditions. This approach simplifies the classification problem by reducing potential overlaps and ambiguities among classes, particularly when certain conditions (e.g., Cyst\_vs\_Normal, Cyst\_vs\_Stone, Cyst\_vs\_Tumor, and Stone\_vs\_Tumor) exhibit subtle variations in imaging features. Additionally, binary classifications allow for optimizing the model's performance for specific diagnostic tasks, ensuring high accuracy and reliability for each condition independently.
>
> 3. ROC curves and AUC metrics were added to provide a more comprehensive analysis of the classification capability, particularly in distinguishing between binary conditions. Including ROC and AUC offers insights beyond standard accuracy, precision, and recall, demonstrating the models' ability to perform consistently across varying thresholds, which is critical for medical diagnostic applications.

---

### Meta-Review · Area_Chair_D5Tt · 2024-12-22

**Metareview:**

This work presents a deep learning framework with transfer learning for the early detection of renal diseases, which outputs four binary classifications: Cyst_vs_Normal, Cyst_vs_Stone, Cyst_vs_Tumor, and Stone_vs_Tumor for a more specific understanding of each stage. By analyzing CT scans and microscopic histopathology images, this work devises convolutional neural networks (CNNs) with pre-trained models to facilitate automatic and precise classification of renal conditions.

**Additional Comments On Reviewer Discussion:**

All four reviewers agree to reject this work, and the rebuttal does not change their ratings. In this regard, this work can not be accepted.

---

### Decision · Program_Chairs · 2025-01-22

Reject